# Targeted suppression of siRNA biogenesis in Arabidopsis pollen promotes triploid seed viability

Kannan Pachamuthu [1,2], Matthieu Simon[1] & Filipe Borges [1] ✉

In plants, small-interfering RNAs (siRNAs) mediate epigenetic silencing via the RNA-directed DNA methylation (RdDM) pathway, which is particularly prominent during reproduction and seed development. However, there is limited understanding of the origins and dynamics of reproductive siRNAs acting in different cellular and developmental contexts. Here, we used the RNaseIII-like protein RTL1 to suppress siRNA biogenesis in Arabidopsis pollen, and found distinct siRNA subsets produced during pollen development. We demonstrate that *RTL1* expression in the late microspore and vegetative cell strongly impairs epigenetic silencing, and resembles RdDM mutants in their ability to bypass interploidy hybridization barriers in the seed. However, germline-specific *RTL1* expression did not impact transgenerational inheritance of triploid seed lethality. These results reveal the existence of multiple siRNA subsets accumulated in mature pollen, and suggest that mobile siRNAs involved in the triploid block are produced in germline precursor cells after meiosis, or in the vegetative cell during pollen mitosis.

Plant small RNA pathways have been extensively studied based on their important roles in plant development and reproduction[1–4]. They are mostly produced as 20- to 24-nucleotides (nt) RNA molecules that are classified as microRNAs (miRNA) and small-interfering RNAs (siRNAs) depending on their precursor transcript and processing machinery[5,6]. The expansion of plant siRNA pathways was propelled by the appearance of RNA polymerases solely dedicated to the production of siRNAs and their targets[5,6], such as Pol IV and Pol V that mediate transcriptional silencing via the RNA-directed DNA methylation (RdDM) pathway[7].

In plants, cytosine methylation is widespread and occurs in the CG, CHG,and CHH contexts (where H is A, C, or T), which are all initiated by RdDM. This pathway is essential for normal development and fertility in many plant species[8–10], but has limited phenotypic impact in the model plant *Arabidopsis thaliana*, which allows studying siRNA biogenesis and activity in different developmental contexts. However, most studies have used genetic mutations that disrupt siRNA biogenesis and activity throughout the entire life cycle, which prevents the identification of siRNA subsets produced in specific tissues and cell types. For example, in the Arabidopsis male gametophyte, three haploid cells differentiate after meiosis to form a mature pollen grain[11]. After the first pollen mitosis, the vegetative cell nucleus exits cell cycle and undergoes epigenetic reprogramming leading to transcriptional activation of transposable elements (TEs) that are rapidly targeted by miRNAs and RdDM[12–16]. In contrast, the generative cell undergoes a second mitosis to generate two sperm cells that progress through S-phase prior to double fertilization, when RdDM pathways are transiently switched off[12,17]. For this reason, epigenetic reprogramming in the male germline has been largely associated to the activity of mobile siRNAs originated from the vegetative cell[14,18,19] and anther tissues during meiosis[20], but the biological significance of this mechanism remains poorly understood.

In order to investigate the origins, dynamics and function of siRNAs accumulated in *A. thaliana* pollen, we developed a method to suppress siRNA biogenesis specifically in pollen by using the antiviral RNaseIII-like protein RTL1 that is able to suppress siRNA biogenesis when expressed by strong constitutive promoters[21]. By using two pollen-specific promoters with very different expression dynamics, we

[1]Université Paris-Saclay, INRAE, AgroParisTech, Institut Jean-Pierre Bourgin (IJPB), Versailles, France. [2]Present address: School of Biosciences and Technology, Vellore Institute of Technology, Vellore, Tamil Nadu, India. ✉e-mail: filipe.borges@inrae.fr

were able to identify distinct TE-derived siRNA subsets produced during pollen development, and demonstrate that RTL1 expression in pollen impairs epigenetic silencing to promote triploid seed viability in the next generation.

## Results

### RTL1-mediated suppression of siRNA biogenesis in Arabidopsis pollen

Ectopic expression of *RTL1* in pollen of Col-0 background was achieved by using the well-known *LAT52* and *MGH3* promoters that are differentially expressed throughout pollen development[22]. While the *LAT52* promoter is expressed in the late microspore stage before becoming preferentially expressed in the vegetative cell during pollen mitosis, the *MGH3* promoter is strongly expressed in the male germline after the first pollen mitosis[22,23]. Ectopic expression of *RTL1* in mature pollen was confirmed in two independent transgenic lines of each construct (Supplementary Fig. 1a), which were subsequently used to perform small RNA sequencing and differential expression analysis (Fig. 1a and Supplementary Fig. 2). Comparative analysis using wild-type pollen (WT Col-0) as control revealed 817 loci with significantly reduced siRNA levels in *pLAT52::RTL1* and 247 in

*pMGH3::RTL1* pollen, which were mostly non-overlapping (Fig. 1b, c, Supplementary Fig. 2a, and Supplementary Data 1). A few hundred loci showing enriched small RNA levels were also detected in both *pLAT52::RTL1* and *pMGH3::RTL1* samples (Supplementary Fig. 2b), but were considered to be indirect normalization effects resulting from the strong loss of siRNA in these lines. Therefore, we focused our analysis on down-regulated siRNA loci that included protein-coding genes targeted by 21/22-nt siRNAs in WT Col-0 pollen. By re-analyzing the transcriptomes of WT Col-0 pollen cell types that were already available[24], we observed that siRNA depletion in *pLAT52::RTL1* pollen corresponded to genes preferentially expressed in the vegetative cell, while reduced 21/22-nt siRNA levels in *pMGH3::RTL1* pollen corresponded to genes preferentially expressed in sperm cells (Fig. 1d and Supplementary Fig. 1c). Taken together, these results confirmed that RTL1 is able to suppress siRNA biogenesis during pollen development, and suggested that the two pollen cell types produce different siRNA subsets.

### RTL1 strongly impairs biogenesis of TE-derived siRNAs in pollen

Interestingly, siRNA depletion in *pLAT52::RTL1* and *pMGH3::RTL1* pollen occurred primarily at TEs that seem to be differentially impacted in

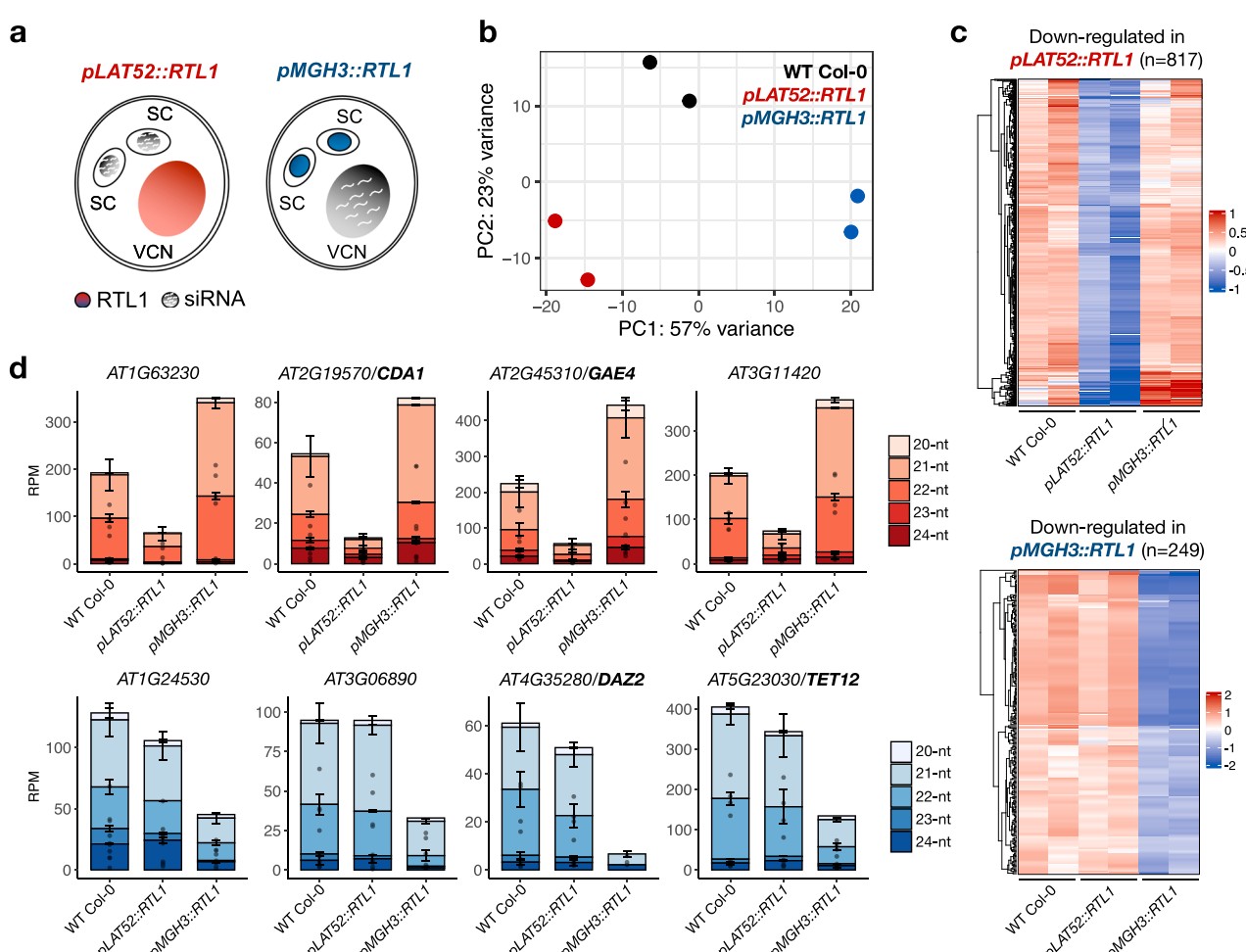

**Fig. 1 | Ectopic expression of *RTL1* suppresses siRNA biogenesis in Arabidopsis pollen. a** Schematic representation of *RTL1* expression in mature pollen driven by the *pLAT52* and *pMGH3* promoters that are preferentially expressed in the vegetative cell and sperm cells, respectively. VCN is vegetative cell nucleus, and SC is sperm cell. **b** Principal component analysis after variance-stabilizing transformations shows reproducibility of small RNA sequencing experiments for two independent transgenic lines of each construct and wild-type Col-0, and highlights variance along PC1 and PC2

between all sample types. **c** Heatmap representation of siRNA depletion in *pLAT52::RTL1* and *pMGH3::RTL1* pollen (vs WT Col-0) shows that distinct loci are affected in the two transgenic lines. **d** Barplots present examples of different protein-coding genes showing reduced levels of 21/22-nt siRNA in *pLAT52::RTL1* or in *pMGH3::RTL1* pollen. Dots and bars represent expression level of individual replicates and mean values (*n* = 2), respectively, error bars represent the standard error, and RPM is reads per million. Source data are provided as a Source Data file.

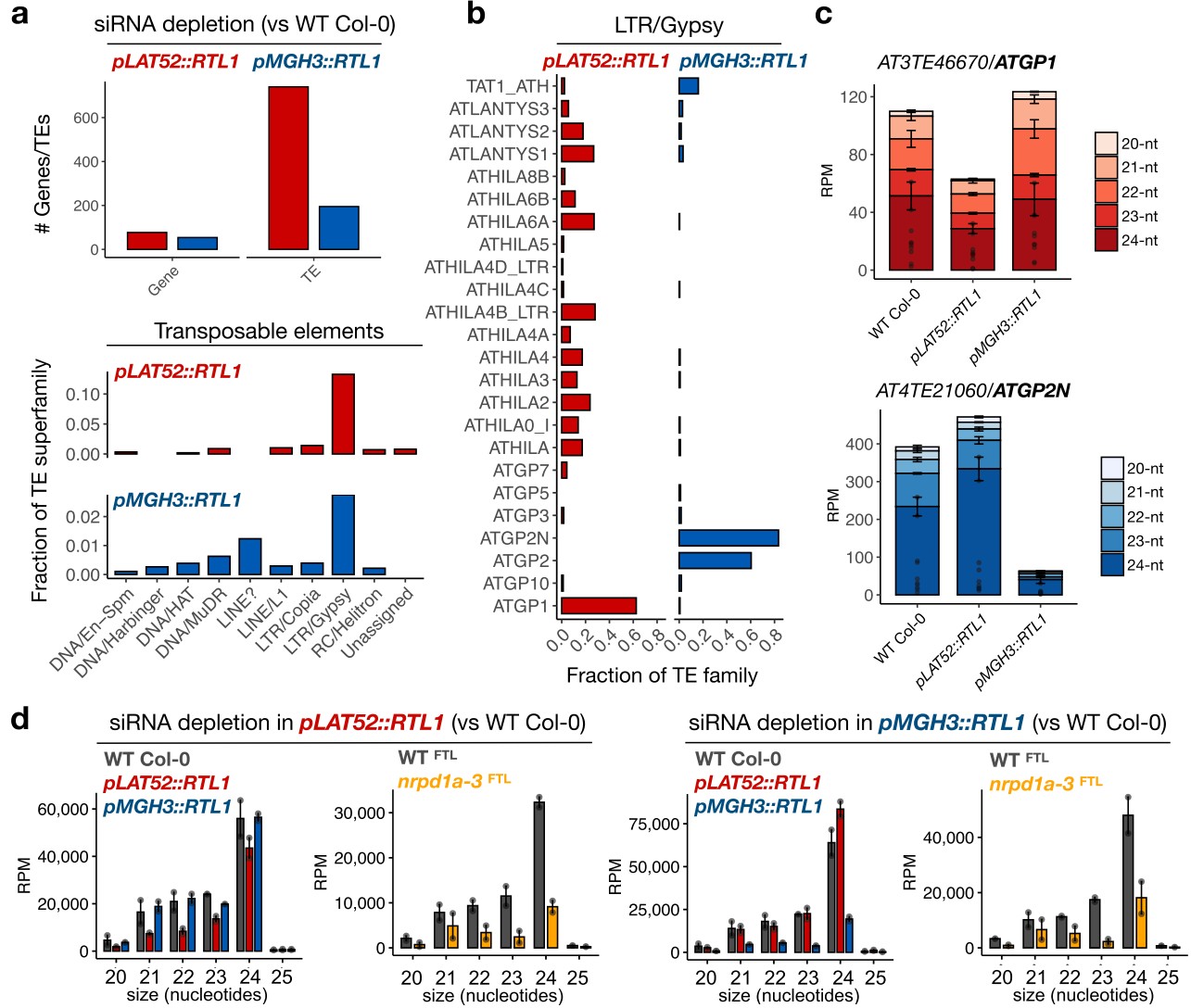

**Fig. 2 | TE-derived siRNAs are differentially expressed in the sperm and vegetative cell lineages via Pol IV. a** Comparative small RNA analysis between transgenic and WT Col-0 pollen shows that siRNA depletion in *pLAT52::RTL1* and *pMGH3::RTL1* pollen occurs primarily at TEs. Among all TEs annotated in the Arabidopsis TAIR10 genome, the LTR/Gypsy superfamily was found over-represented in the lists of differentially expressed siRNA loci. **b** Different members of the LTR/Gypsy superfamily showed significantly reduced siRNA levels in *pLAT52::RTL1* or *pMGH3::RTL1* pollen. The bar plots show the fraction of TE

superfamilies in the two datasets. **c** Many retrotransposons from the *ATGP1* family lost siRNAs specifically in *pLAT52::RTL1* pollen, while *ATGP2* and *ATGP2N* lost siRNAs specifically in *pMGH3::RTL1* pollen. The bar plots show the fraction of TE families in the two datasets. **d** siRNAs depleted in *pLAT52::RTL1* or *pMGH3::RTL1* pollen are dependent on Pol IV activity. Dots and bars represent expression level of individual replicates and mean values (*n* = 2), respectively, error bars represent the standard error, and RPM is reads per million. Source data are provided as a Source Data file.

the two transgenic lines (Fig. 2a–c). This includes *ATGP1, ATHILA4, VANDAL3* and *ATLANTYS2* families that lost siRNAs mainly in *pLAT52::RTL1* pollen, while siRNAs from *ATGP2, ATGP2N, VANDAL6* and *ATCOPIA36* families were primarily depleted in *pMGH3::RTL1* pollen (Fig. 2b, c and Supplementary Fig. 3). Depletion of 24-nt siRNAs in *pMGH3::RTL1* pollen was particularly surprising, given that downstream RdDM components are not expressed in sperm cells[12,25]. To confirm that these siRNAs are indeed produced within the canonical RdDM pathway during pollen development, we took advantage of a Fluorescence Tagged Line (FTL1230)[26] carrying a pollen-expressed DsRed transgene genetically linked to *NRPD1a* (largest subunit of Pol IV) (Supplementary Fig. 4a). This FTL line was crossed with the *nrpd1a-3* mutant to obtain double heterozygous FTL1230/+;*nrpd1a-3*/+ plants, which allowed purification of segregating wild-type (DsRed positive, WT[FTL]) and mutant (DsRed negative, *nrpd1a-3*[FTL]) pollen by fluorescence-activated cell sorting (FACS) (Supplementary Fig. 4b, c). This experiment showed that most TEs within the LTR/Gypsy

superfamily lost siRNA in *nrpd1a-3*[FTL] pollen, including the *ATGP1* and *ATGP2N* families that were the most affected in *pLAT52::RTL1* and *pMGH3::RTL1* pollen, respectively (Fig. 2c, d and Supplementary Fig. 4d). These results demonstrate that the multiple TE-derived siRNA subsets depleted in *pLAT52::RTL1* and *pMGH3::RTL1* pollen are at least partially dependent on the gametophytic activity of Pol IV, which is the main pathway for siRNA biogenesis in Arabidopsis pollen[15,27,28].

## RTL1 expression impacts the pollen epigenome

RTL1-mediated suppression of siRNA biogenesis specifically in pollen provided an opportunity to investigate the role of gametophytic siRNAs in germline reprogramming and epigenetic inheritance. Therefore, we profiled DNA methylation in *pLAT52::RTL1* and *pMGH3::RTL1* pollen and seedlings by whole-genome bisulfite sequencing (WGBS), which was compared to WT Col-0, FACS-purified WT[FTL] and *nrpd1a-3*[FTL] pollen, as well as *nrpd1a* and *p35S::RTL1* seedlings as controls (Supplemental Data 5). The analysis of differentially methylated regions

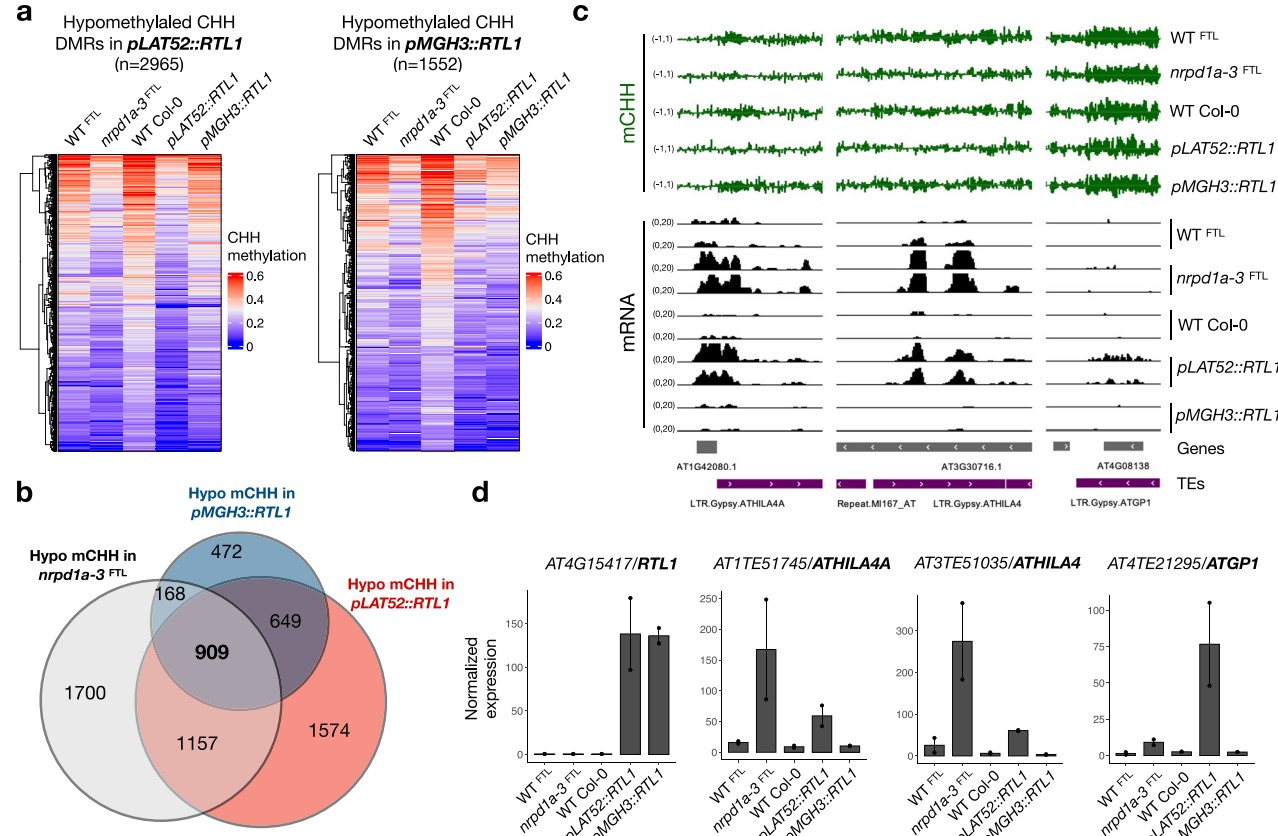

**Fig. 3 | RTL1-mediated suppression of siRNA biogenesis impacts the pollen epigenome. a** Heatmap and boxplot representation of CHH methylation levels at hypomethylated DMRs in *pLAT52::RTL1* and *pMGH3::RTL1* pollen (as compared to WT Col-0) shows overlap with reduced CHH levels in *nrpd1a-3* FTL pollen. **b** Venn diagram shows the overlaps between hypomethylated regions in CHH context in *nrpd1a-3* FTL, *pLAT52::RTL1* and *pMGH3::RTL1* pollen. The statistical significance of each overlap was calculated using the R package SuperExactTest[48]. **c** Genome browser tracks show CHH methylation (green) and RNA sequencing (black) data-sets at selected loci. Partial CHH hypomethylation is observed specifically in

*nrpd1a-3* FTL and *pLAT52::RTL1* pollen, leading to the transcriptional activation of retrotransposons. **d** RNA sequencing confirmed *RTL1* expression in both *pLAT52::RTL1* and *pMGH3::RTL1* pollen, but up-regulation of LTR/Gypsy retro-elements from the *ATHILA4A*, *ATHILA4* and *ATGP1* families occurs only in *pLAT52::RTL1* and *nrpd1a-3* FTL pollen. Dots and bars represent expression level of individual replicates and mean values (*n* = 2), respectively, error bars represent the standard error, and normalized expression values were calculated by DESeq2 using the median of ratios method. Source data are provided as a Source Data file and in Supplementary Data 2.

(DMRs) in *pLAT52::RTL1* and *pMGH3::RTL1* pollen detected 2965 and 1552 hypomethylated DMRs in the CHH context, respectively (Fig. 3a), which mapped mainly to TEs and intergenic regions that were similarly hypomethylated in *nrpd1a-3*FTL pollen (Fig. 3b, Supplementary Fig. 6a, and Supplementary Data 2). Combined, the methylomes of *pLAT52::RTL1* and *pMGH3::RTL1* pollen resembled that of *nrpd1a-3*FTL to a large extent (Fig. 3b and Supplementary Fig. 5), although differences were expected given that RTL1 targets all siRNA types[21]. In contrast, the methylomes of *pLAT52::RTL1* and *pMGH3::RTL1* seedlings showed only a small number of hypo- and hypermethylated CHH DMRs as compared to *p35S::RTL1* and *nrpd1a* seedlings (Supplementary Figs. 6a and 7), suggesting that the loss of pollen siRNAs has a limited impact on RdDM activity in the sporophyte. Importantly, the striking differences in siRNA depletion between *pLAT52::RTL1* and *pMGH3::RTL1* pollen were not reflected in their methylomes, as we found a significant overlap between hypomethylated regions in the two datasets (Fig. 3b and Supplementary Fig. 6b, c). This could be partially explained by the poor representation *ATGP2* and *ATGP2N* families in the list of hypomethylated DMRs (Supplementary Fig. 6c), while these two families represent the vast majority of TEs that lost siRNA in *pMGH3::RTL1* and *nrpd1a-3*FTL pollen (Fig. 2b and Supplementary Fig. 4d). This result suggests that *ATGP2* and *ATGP2N* siRNAs are not actively involved in RdDM in pollen. Indeed, transcriptome analysis by mRNA sequencing showed that TEs remained

transcriptionally repressed in *pMGH3::RTL1* pollen, as only six genes (including *RTL1*) were significantly deregulated (Supplementary Fig. 8 and Supplementary Data 3). In contrast, the transcriptome of *pLAT52::RTL1* pollen showed 718 up-regulated and 855 down-regulated genes and TEs (Supplementary Fig. 8 and Supplementary Data 3), including retrotransposons from the *ATHILA4A*, *ATHILA4*, and *ATGP1* families that were equally up-regulated in *nrpd1a-3*FTL mutant pollen (Fig. 3c, d). These analyses demonstrate that *RTL1* expression driven by the LAT52 promoter strongly impairs epigenetic silencing during pollen development, but has a limited impact on RdDM activity in the next generation.

## RTL1 expression in pollen impacts triploid seed viability
Previous studies have shown that the paternal RdDM pathway triggers interploidy hybridization barriers in the seed, which is known as the "triploid block" response[15,27,29,30]. Therefore, to test if RTL1-mediated suppression of pollen siRNAs also allows bypassing the triploid block, we introduced *pLAT52::RTL1* and *pMGH3::RTL1* transgenes into the *jas-3* mutant (Col-0 background) that produces 30 to 40% of unreduced diploid pollen and triploid seeds that collapse at high frequencies[31,32]. Strikingly, a significant decrease in triploid seed collapse was observed in three independent *jas-3;pLAT52::RTL1* lines over five consecutive generations, while there was no significant effect in *jas-3;pMGH3::RTL1* plants (Fig. 4a, b and Supplementary Fig. 9a). Importantly, the lower

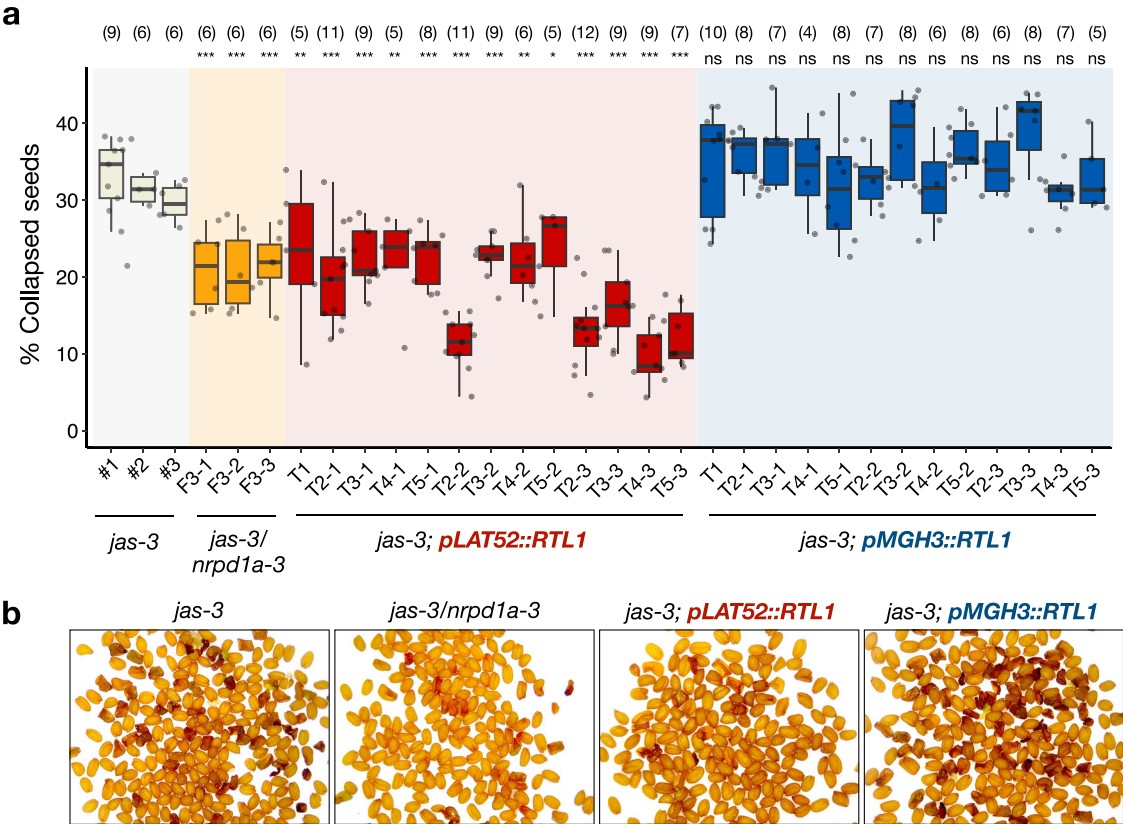

**Fig. 4 | The triploid block is bypassed only when RTL1 is preferentially expressed in the vegetative cell. a** The triploid block response in *jas-3* mutants is quantified by counting the number of collapsed seeds in six siliques of selfed plants, which reflects the amount of diploid pollen produced in this mutant background (30 to 40%). Seed abortion is significantly reduced in *jas-3/nrpd1a-3* double mutant plants in the F₃ generation, and also across five consecutive generations of three independent *jas-3;pLAT52::RTL1* lines, while *jas-3;pMGH3::RTL1* remains similar to *jas-3* controls. Numbers above each box represent the number of individual plants used for each genotype, and their levels of triploid seed collapse is represented by the dots on top of each box. Boxes represent the interquartile range (IQR) showing the lower (Q1) and upper (Q3) quartiles surrounding the median (central line), and whiskers represent the minimum (Q1 −1.5*IQR) and maximum (Q3 + 1.5*IQR) values. Statistically significant differences in the percentage of collapsed seeds were calculated by ANOVA with a post hoc Dunnett test, using *jas-3* as the reference group (ns is not significant, * is *P* < 0.05, ** is *P* < 0.01, *** is *P* < 0.001). Source data are provided as a Source Data file. **b** Representative images of seeds show lower levels of seed abortion in suppressor lines *jas-3/nrpd1a-3* and *jas-3;pLAT52::RTL1*, in comparison with a non-suppressor line *jas-3;pMGH3::RTL1* and *jas-3* controls.

levels of triploid seed collapse in *jas-3;pLAT52::RTL1* plants resembled the *jas-3;nrpd1a-3* double mutant (Fig. 4b), suggesting that the suppressive effect is caused by the loss of Pol IV-dependent siRNAs that have been previously implicated in the triploid block response[27,29,30]. However, we cannot discard the possibility that other siRNA types are also involved, or that the suppressive effect mediated by *RTL1* expression in pollen is not associated to siRNA activity. Importantly, increased viability of triploid seeds in *jas-3;pLAT52::RTL1* lines coincided with down-regulation of the paternally expressed imprinted gene *PHERES1* (*PHE1*) in developing siliques, which is an essential regulator of genomic imprinting and triploid seed lethality in Arabidopsis[33] (Supplementary Fig. 9b). This demonstrates that *RTL1* expression during pollen development impacts imprinted gene expression in interploid hybrid seeds with paternal excess, thus mimicking strong epigenetic suppressors of the triploid block[27,34,35].

## Discussion

We developed a method to suppress siRNA biogenesis in Arabidopsis pollen, based on the ectopic expression of *RTL1* driven by two pollen-specific promoters. Our results provide conclusive evidence that siRNA biogenesis is reinforced during pollen development, including the male germline where RdDM activity is significantly reduced[12,13,25]. Indeed, RTL1 targeted a completely different subset of TE siRNAs when expressed by the germline-specific promoter *MGH3*, perhaps reflecting the distinctive chromatin organization of this cell type[12,13,36−38]. This highlights the importance of having mobile siRNAs produced in germline precursor cells or companion vegetative cells in pollen[14,18−20,23], and in the female gametophyte[39], as it may compensate the fact that siRNA biogenesis in the germline is limited to only certain TEs. Our results provide further support to these ideas, as *RTL1* expression driven by the *LAT52* promoter allowed bypassing the triploid block in hybrid seeds, while germline-specific *RTL1* expression did not impact triploid seed lethality. This strongly suggests the existence of mobile siRNAs originated from the late microspore or vegetative nucleus, which must then accumulate in the germline in order to reach the endosperm after fertilization, possibly loaded in some of the many Argonaute (AGO) proteins that are highly abundant in sperm cells[40]. AGO1 and AGO5 are good candidates in this model, as they have been both independently implicated in the triploid block response[16]. This hypothesis would also explain why RTL1 activity in the germline is not able to target siRNAs originated from elsewhere, as mobile siRNAs loaded into AGO proteins may be protected from RTL1 activity. Alternatively, mobile siRNAs accumulated in sperm cells could be modified in a way that prevents targeting by RTL1. Indeed, pollen siRNAs involved the triploid block were recently shown to be heavily modified by pseudouridine, which seems to be an essential feature of mobile siRNAs in both plants and mammals[41].

The biological significance of active siRNA biogenesis in sperm cells remains unclear, given that hundreds of genes and TEs produce siRNAs that are seemingly not engaged in PTGS and RdDM activity in the male germline. This is particularly puzzling for *ATGP2* and *ATGP2N* siRNAs, which combined represent a large fraction within the most abundant TE-derived siRNAs detected in wild-type Arabidopsis pollen (Supplementary Fig. 3c). As downstream components of RdDM are not expressed in sperm cells[12], one possibility is that sperm-borne siRNAs are sequestered only to participate in RdDM later in germline specification during pollen tube growth, or maybe after fertilization in the early seed. A recent analysis of parental siRNA contributions in the Arabidopsis endosperm supports this idea, as the loss of paternal siRNAs resulted in reduced DNA methylation and significant changes in gene expression[42]. However, our results have shown that *RTL1* expression in pollen had only a minor impact on RdDM activity in the sporophyte (Supplementary Fig. 6), suggesting that siRNA biogenesis in pollen is mostly dispensable for transgenerational epigenetic inheritance. This raises yet another possibility that sperm-borne small RNAs are mainly involved in translational repression, as previously shown with artificial miRNAs[23]. It is tempting to speculate that a subset of germline small RNAs is part of a surveillance mechanism that is primed to act only in situations of epigenetic instability caused by stress or hybridizations that lead to bursts of TE expression and activity. These outstanding questions may now be tested by using RTL1-mediated suppression of sperm siRNAs in different plant species and developmental contexts.

## Methods

### Plant materials, plasmid cloning, and plant transformation

The Arabidopsis mutant lines *jas-3* (SAIL_813_H03) and *nrpd1a-3* (SALK_128428), as well as the fluorescence-tagged line FTL1230[26] were used in this study, and are all in Col-0 background. Plants were grown in greenhouse long-day conditions (16 h light and 8 h dark). The *pLAT52::RTL1-Myc* and *pMGH3::RTL1-Myc* constructs were generated by PCR amplification of the respective promoters from genomic DNA (primers listed in Supplementary Table 1), and cloned into the *p35S::RTL1-Myc* binary plasmid[21]. The 35S promoter was replaced using *Hind*III and *Xba*I restriction sites, and each plasmid was transformed into Arabidopsis Col-0 and *jas-3* mutants by floral dipping[43]. Transgenic seeds were surface sterilized with 50% bleach followed by 70% ethanol for 2 min, washed with sterile deionized water and sowed on agar plates (0.5X MS medium, 1% sucrose, pH 5.7) supplemented with cefotaxime (250 mg/L, Duchefa) and hygromycin (25 mg/L, Duchefa). Plates were placed in a growth chamber at 23 °C, 70% humidity, 120 µE m$^{-2}$ light with a 16-h light/8-h dark (long days) photoperiod for two weeks, and seedlings were then transferred to soil and grown in greenhouse long-day conditions to complete the life cycle (16 h light and 8 h dark).

### RT-qPCR analysis

Pollen was purified by collecting open flowers into a 50 mL falcon tube, vortexing for 3 min in 20 mL of 100 mM phosphate buffer, and filtering through a 50 µm mesh. Pollen was then concentrated by centrifugation (3 min, 5000 RPM), and disrupted by shacking in the presence of glass beads (Sigma) with a Retsch homogenizer. Total RNA was then extracted using the Direct-zol RNA Microprep kit (Zymo) following the manufacturer's instructions. Developing siliques were collected 7 days after pollination, and total RNA was extracted using the RNeasy Plant Mini Kit (Qiagen) following the manufacturer's recommendations for seed tissues (RLC buffer). 1 µg of total RNA was then subjected to DNase (Invitrogen) treatment and converted into cDNA using Superscript II RT and random primers (Invitrogen). RT-qPCR was performed on a CFX Connect Real-Time PCR machine (BioRad) using SsoAdvanced Universal SYBR Green Supermix (BioRad). Primers used for RT-qPCR reaction are listed in Supplementary Table 1. *MGH3* and *ACT2* were used as internal controls.

### Fluorescence-activated cell sorting

The FTL1230 line contains a pollen-expressed transgene genetically linked to *NRPD1a* (Supplementary Fig. 6), and is composed of a DsRed marker gene driven by the *LAT52* promoter[26]. This line was crossed with the homozygous mutant *nrpd1a-3*, and the resulting heterozygous *nrpd1a-3/+* F1 plants allowed purification of wild-type (DsRed positive) and mutant *nrpd1a-3* (DsRed negative) pollen by Fluorescence-Activated Cell Sorting (FACS) (Supplementary Fig. 6). Pollen was purified by collecting open flowers into Eppendorf tubes, vortexing in 2 mL of 100-mM sodium phosphate buffer (pH 7) for 3 min, and filtering through a 50 µm nylon mesh (CellTrics, Sysmex). Pollen was purified using a MoFlo Astrios EQ cell sorter (Beckman-Coulter) equipped with a 561 nm laser (200 mW) for DsRed excitation. Pollen populations are characterized by an elevated high angle scatter (SSC) and auto-fluorescence, which combined with the differences in DsRed signal intensity allowed to sort WT$^{FTL}$ (DsRed +) and *nrpd1a-3*$^{FTL}$ (DsRed -) pollen simultaneously (Supplementary Fig. 4c). Approximately 500,000 pollen grains of each population were used for downstream analysis.

### Small RNA sequencing and analysis

Library construction and sequencing was performed at BGI Genomics (Hong Kong). Single-end 50-nt reads were pre-processed by filtering collapsed reads according to length and quality. Filtered reads were mapped to the Arabidopsis TAIR10 genome with bowtie[44], reporting all multi-mappers. Only perfect-match reads were used for all downstream analysis. Reads were normalized by dividing non-redundant read counts by the number of genomic hits, and subsequently calculating the number of reads per million of filtered (18–30 nt) and perfectly mapped reads. Differential expression analysis was performed with the R package DESeq2[45] using an FDR adjusted *p*-value of 0.05. Additional downstream analyses and plots were done with custom R scripts. Graphical outputs were produced using the R package "ggplot2". A summary of all small-RNA sequencing data generated in this study is presented in Supplementary Data 4.

### Whole-genome bisulfite sequencing and DNA methylation analysis

Genomic DNA was purified from total pollen and two-week-old seedlings, and library preparation and sequencing were performed by BGI Genomics (Hong Kong) as paired-end 100 bp reads using DNBSEQ technology. Pre-processed and high-quality reads were mapped to the TAIR10 genome using bismark with default settings for paired-end libraries[46], and all figures and downstream analysis were performed using custom R scripts. Graphical outputs were produced using the R package "ggplot2" and "ComplexHeatmap". Methylome data from *nrpd1a-3* seedlings was already available[47]. DMRs were defined as 100-bp bins containing at least 4, 5, or 6 differentially methylated CGs, CHGs, or CHH and with an absolute methylation difference of at least 0.4, 0.2, or 0.1, respectively. Bins localizing within 200 bp of each other were merged and considered as DMRs. The statistical significance of the observed overlaps between DMRs was calculated using the R package "SuperExactTest"[48], and the analysis is presented in Supplementary Data 5. A summary of all WGBS sequencing data generated in this study is presented in Supplementary Data 4.

### RNA sequencing and analysis

Sequencing of messenger RNA was performed by BGI Genomics (Hong Kong) as paired-end 100 bp reads using DNBSEQ technology. High-quality raw reads were aligned to the TAIR10 genome using STAR[49]. Normalization and analysis of differential gene expression was performed using the R package DESeq2[45], with an FDR adjusted *p*-value of 0.05. Graphical outputs were produced using the R packages "ggplot2", "pheatmap", and "ComplexHeatmap". A summary of all RNA sequencing data generated in this study is presented in Supplementary Data 4.

## Triploid block quantification and statistical analysis

The triploid block response in *jas-3* lines was quantified by imaging dry seeds from six siliques under a stereoscopic microscope (Nikon), and counting the number of normal and collapsed seeds. Statistically significant differences in the percentage of collapsed seeds were calculated by one-way analysis of variance (ANOVA) with a post hoc Dunnett test, using the R packages "ggpubr" and "multcomp". These results are presented in Supplementary Data 5.

## Reporting summary

Further information on research design is available in the Nature Portfolio Reporting Summary linked to this article.

## Data availability

All sequencing datasets generated in this study are publicly available in the NCBI's Gene Expression Omnibus under the accession number GSE231670. Expression profiling of Col-0 pollen nuclei is available in GSE155369. Source data are provided with this paper.

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

## Acknowledgements

This work was supported by the grant EpiHYBRIDS from the French National Agency of Research (ANR-19-CE120008 to F.B.). We thank all members of the EPIREP team for daily discussions, Hervé Vaucheret for critical reading of the manuscript and sharing the *p35S::RTL1-Myc* plasmid and seeds, and Gregory Copenhaver for the FTL1230 line. The authors acknowledge technical support from Mickael Bourge and Nicolas Valentin from the cytometry facility of the I2BC. This work has benefited from the support of IJPB's Plant Observatory technological platforms. The IJPB benefits from the support of Saclay Plant Sciences-SPS (ANR-17-EUR-0007).

## Author contributions

F.B. designed the study; K.P. and M.S. performed the experiments; F.B. analyzed the data and wrote the manuscript with contributions from K.P. and M.S.

## Competing interests

The authors declare no competing interests.
