## [Peer Review File · Nature Communications]

Targeted suppression of siRNA biogenesis in Arabidopsis pollen promotes triploid seed viabilityEditorial Note: This manuscript has been previously reviewed at another journal that is not operating a transparent peer review scheme. This document only contains reviewer comments and rebuttal letters for versions considered at *Nature Communications*.

REVIEWER COMMENTS

Reviewer #1 (Remarks to authors):

This is an interesting manuscript and body of work. However, there seems to be a controversy from the previous rounds of review regarding the use of the pollen cell-type-specific promoters. This manuscript repeatedly makes claims regarding sperm cell vs. vegetative cell specificity based on their results, yet at the same time does not individually isolate and test these different cell types, and has used a promoter that is not ideal because it is known to be not fully specific to one of the two cell types. Therefore, I see the following below as a compromising solution.

1. The authors need to be up-front with the reader about the cell-specificity of the Lat52 promoter. When they write that Lat52 is preferentially expressed in this cell type, they are not being clear to the reader about what previous papers have shown. Therefore, they should be clear that previous reports demonstrate Lat52-expressed transcripts accumulating in the precursor to the sperm cells.
2. Later in the manuscript the Lat52 promoter is again used to drive the NRPD1a transcript, although at this point it is called a “pollen promoter”. And at this point in the manuscript the results show a change in ATGP2N, which is thought to be a sperm-specific transcript. Again, the authors need to be clear to the reader about where exactly this transcript is expressed from.
3. Taken #1 and #2 into account, the authors should reduce their assumptions and claims that their data shows cell-specific results throughout the manuscript, including many times in the Abstract. If the authors would like to speculate on the cell specificity, they should clearly mark it as speculation and that a particular event is “thought to occur in” or is

“presumably in the sperm cells”. Otherwise the authors should use the term “pollen” as they have in the title as not to make a claim about cell-type specificity. There are many instances of this throughout the manuscript.

Reviewer #2 (Remarks to authors):

I thank the authors for revising their manuscript and further explaining their points. I agree with the authors that whether the small RNAs that repress the triploid block are from microspores or vegetative cells does not necessarily change the impact of their research. However, the movement from VC to SC was what is written in the abstract and text of the previous and revised manuscripts. In my opinion, discussing the alternative possibility that the small RNAs responsible for the triploid block are from microspores would improve the manuscript. Discussing both models may even increase the impact of their manuscript. Related to this, the Grant-Downton work argued against the evidence for small RNA movement between the VN and SC published by Slotkin et al. The lack of movement could explain why artificial miRNAs against GFP under the control of the VCK promoter were not able to repress germline GFP. Artificial miRNAs under the control of the LAT52 promoter were able to repress germline GFP presumably because they were expressed in the late microspore and generative cells.

We thank again the two reviewers for carefully reading the different versions of our manuscript, and the helpful suggestions to improve it further. Our detailed responses are listed below.

Reviewer #1 (Remarks to authors):

This is an interesting manuscript and body of work. However, there seems to be a controversy from the previous rounds of review regarding the use of the pollen cell-type-specific promoters. This manuscript repeatedly makes claims regarding sperm cell vs. vegetative cell specificity based on their results, yet at the same time does not individually isolate and test these different cell types, and has used a promoter that is not ideal because it is known to be not fully specific to one of the two cell types. Therefore, I see the following below as a compromising solution.

1. The authors need to be up-front with the reader about the cell-specificity of the Lat52 promoter. When they write that Lat52 is preferentially expressed in this cell type, they are not being clear to the reader about what previous papers have shown. Therefore, they should be clear that previous reports demonstrate Lat52-expressed transcripts accumulating in the precursor to the sperm cells.
2. Later in the manuscript the Lat52 promoter is again used to drive the *NRPD1a* transcript, although at this point it is called a “pollen promoter”. And at this point in the manuscript the results show a change in *ATGP2N*, which is thought to be a sperm-specific transcript. Again, the authors need to be clear to the reader about where exactly this transcript is expressed from.
3. Taken #1 and #2 into account, the authors should reduce their assumptions and claims that their data shows cell-specific results throughout the manuscript, including many times in the Abstract. If the authors would like to speculate on the cell specificity, they should clearly mark it as speculation and that a particular event is “thought to occur in” or is “presumably in the sperm cells”. Otherwise the authors should use the term “pollen” as they have in the title as not to make a claim about cell-type specificity. There are many instances of this throughout the manuscript.

We agree that there were still many instances where the *pLAT52::RTL1* construct was only referring to *RTL1* expression in the vegetative nucleus. Therefore, we have revised the text exhaustively to clarify the different possibilities associated to the *LAT52* promoter, and moved the speculation as much as possible to the discussion section. Please see highlighted text throughout the revised manuscript.

In response to point #2, we did not use the *LAT52* promoter to express *NRPD1a*. This experiment used the *FTL1230* line that carries a transgene containing the *LAT52* promoter driving expression of the fluorescent protein DsRed. This transgene is genetically linked to *NRPD1a* (Supplementary Fig. 4). Therefore, when the *FTL1230* line was crossed with the *nprp1a-3* homozygous mutant, we were able to use F1 double heterozygous plants to purify wild-type (DsRed +) and *nprp1a-3* mutant pollen (DsRed -) by FACS. This allowed to confirm that siRNAs depleted in *pLAT52::RTL1* and *pMGH3::RTL1* lines are indeed produced after meiosis, and are mostly dependent on the gametophytic activity of Pol IV.

Reviewer #2 (Remarks to authors):

I thank the authors for revising their manuscript and further explaining their points. I agree with the authors that whether the small RNAs that repress the triploid block are from microspores or vegetative cells does not necessarily change the impact of their research. However, the movement from VC to SC was what is written in the abstract and text of the previous and revised manuscripts. In my opinion, discussing the alternative possibility that the small RNAs responsible for the triploid block are from microspores would improve the manuscript. Discussing both models may even increase the impact of their manuscript. Related to this, the Grant-Downton work argued against the evidence for small RNA movement between the VN and SC published by Slotkin et al. The lack of movement could explain why artificial miRNAs against GFP under the control of the VCK promoter were not able to repress germline GFP. Artificial miRNAs under the control of the LAT52 promoter were able to repress germline GFP presumably because they were expressed in the late microspore and generative cells.

We have now addressed all these important points, and have modified substantially the abstract and discussion sections to better reflect the different scenarios. Please see highlighted text in the revised manuscript.

REVIEWERS' COMMENTS

Reviewer #1 (Remarks to the Author):

The authors have done a nice job revising the manuscript and removing claims that they can't be 100% certain of due to the early pollen expression of the Lat52 promoter.

In the Results section titled "RTL1 strongly impairs biogenesis of TE-derived siRNAs in pollen", the last three sentences detail an experiment and data that demonstrates that TE siRNAs in pollen are generated by RNA Polymerase IV. This is stated as if it is new data, but it actually has been known for a while and references should be added here. For example: PMID 32075560.

Reviewer #2 (Remarks to the Author):

The authors have addressed my concerns, and I have no additional comments. In my opinion, this manuscript is ready for publication.

Reviewers' Comments:

Reviewer #1 (Remarks to the Author):

The authors have done a nice job revising the manuscript and removing claims that they can't be 100% certain of due to the early pollen expression of the Lat52 promoter.

In the Results section titled "RTL1 strongly impairs biogenesis of TE-derived siRNAs in pollen", the last three sentences detail and experiment and data that demonstrates that TE siRNAs in pollen are generated by RNA Polymerase IV. This is stated as if it is new data, but it actually has been known for a while and references should be added here. For example: PMID 32075560.

Thanks! We agree and modified the sentence to clarify this important point.

These results demonstrate that the multiple TE-derived siRNA subsets depleted in *pLAT52::RTL1* and *pMGH3::RTL1* pollen are at least partially dependent on the gametophytic activity of Pol IV, which is the main pathway for siRNA biogenesis in *Arabidopsis* pollen^{15,27,28}.

Reviewer #2 (Remarks to the Author):

The authors have addressed my concerns, and I have no additional comments. In my opinion, this manuscript is ready for publication.

We thank again the two reviewers for their positive comments and suggestions that certainly helped to improve significantly our manuscript.